# Radiotracers for Bone Marrow Infection Imaging

**DOI:** 10.3390/molecules26113159

**Published:** 2021-05-25

**Authors:** Lars Jødal, Pia Afzelius, Aage Kristian Olsen Alstrup, Svend Borup Jensen

**Affiliations:** 1Department of Nuclear Medicine, Aalborg University Hospital, DK-9000 Aalborg, Denmark; svbj@rn.dk; 2Zealand Hospital, Køge, Copenhagen University Hospital, DK-4600 Køge, Denmark; pafz@regionsjaelland.dk; 3Department of Nuclear Medicine & PET, Aarhus University Hospital, DK-8200 Aarhus, Denmark; aagols@rm.dk; 4Department of Clinical Medicine, Aarhus University, DK-8000 Aarhus, Denmark; 5Department of Chemistry and Biosciences, Aalborg University, DK-9220 Aalborg, Denmark

**Keywords:** osteomyelitis, radionuclides, SPECT, PET, Ga-citrate, IL-8, FDG, FDS, Siglec-9, methionine, ubiquicidin, UBI

## Abstract

Introduction: Radiotracers are widely used in medical imaging, using techniques of gamma-camera imaging (scintigraphy and SPECT) or positron emission tomography (PET). In bone marrow infection, there is no single routine test available that can detect infection with sufficiently high diagnostic accuracy. Here, we review radiotracers used for imaging of bone marrow infection, also known as osteomyelitis, with a focus on why these molecules are relevant for the task, based on their physiological uptake mechanisms. The review comprises [^67^Ga]Ga-citrate, radiolabelled leukocytes, radiolabelled nanocolloids (bone marrow) and radiolabelled phosphonates (bone structure), and [^18^F]FDG as established radiotracers for bone marrow infection imaging. Tracers that are under development or testing for this purpose include [^68^Ga]Ga-citrate, [^18^F]FDG, [^18^F]FDS and other non-glucose sugar analogues, [^15^O]water, [^11^C]methionine, [^11^C]donepezil, [^99m^Tc]Tc-IL-8, [^68^Ga]Ga-Siglec-9, phage-display selected peptides, and the antimicrobial peptide [^99m^Tc]Tc-UBI_29-41_ or [^68^Ga]Ga-NOTA-UBI_29-41_. Conclusion: Molecular radiotracers allow studies of physiological processes such as infection. None of the reviewed molecules are ideal for the imaging of infections, whether bone marrow or otherwise, but each can give information about a separate aspect such as physiology or biochemistry. Knowledge of uptake mechanisms, pitfalls, and challenges is useful in both the use and development of medically relevant radioactive tracers.

## 1. Introduction

The application of radioactive molecules allows the study of the physiology of the human body, and they are regularly used in hospitals for the diagnosis of a variety of diseases using positron emission tomography (PET) or gamma-camera imaging. This review will focus on radioactive tracers (radiotracers) for bone marrow infection. 

Bone marrow infection can turn into osteomyelitis, an inflammatory process accompanied by bone destruction [1]. Osteomyelitis can be caused by microorganisms introduced directly into the bone, e.g., from a bone fracture with skin puncture or from bacterial contamination during bone surgery. Alternatively, the infection can be blood-borne (haematogenous), with microorganisms reaching the blood through an (untreated) wound and thereby reaching the bone. On a global scale, the majority of osteomyelitis cases are among children, especially in low-income countries, and musculoskeletal impairment due to infection has been estimated to affect 12 million (3%) children in the least developed countries [2]. Prompt diagnosis and therapy are of particular importance in cases of aggressive and multifocal disease. X-ray imaging can be used for diagnosis, but by nature, X-rays cannot visualise osteomyelitis until the bone matrix has been affected enough to make a difference in X-ray attenuation. 

Radiotracer imaging, on the other hand, can be used to study the processes leading to destruction rather than waiting for the destruction to happen. An important aspect that distinguishes body physiology from chemical reactions in a test tube is that the body is inhomogeneous, consisting of tissues that have a different interaction with the same substance. More specifically, some substances will have a higher uptake in sick bone than in healthy bone (or vice versa). If the substance contains a radionuclide, its uptake can be traced by radiation-sensitive imaging apparatus such as a gamma camera or a PET scanner. In general, the use of short-lived radionuclides (half-lives from days down to minutes) can ensure that the patient radiation dose is comparable to or lower than the radiation dose from a CT scan.

As an example, the element gallium concentrates in infected areas, making the radioisotopes ^67^Ga and ^68^Ga possible tracers for the imaging of osteomyelitis (further details follow below). Another radiotracer approach is to synthesize a molecule with physiology that makes it concentrate in the infected tissue, a molecule that contains or carries a radionuclide suitable for imaging. An important example is fluoro-deoxy-glucose (FDG), whose uptake in tissue mimics the uptake of glucose. This is relevant as infected tissue has increased glucose uptake. Accordingly, the radiotracer [^18^F]FDG can be used for infection imaging. 

In some radiotracers, such as [^15^O]water = [^15^O]H_2_O, the radionuclide is an integrated element of the molecule as a whole. In other cases, such as [^18^F]FDG, the physiology is determined by the rest of the molecule—the uptake of deoxy-glucose is similar to the uptake of glucose, while fluorine has no active chemical role in the uptake, it just does not disturb the process (further details later). The principle of labelling a larger unit with interesting physiology can also be used on the body’s own cells. Leukocytes or white blood cells are an integrated part of the body’s defence mechanisms against infection, for which reason leukocytes naturally accumulate in infected tissues. Labelling of leukocytes with a radionuclide, such as ^111^In or ^99m^Tc, allows imaging of the leukocyte uptake in infection.

Radiotracer imaging of infection goes back half a century [3], and several radiotracers are routinely used in osteomyelitis diagnostics. However, no radiotracers are perfect. Some tracers have physiological uptake in healthy tissue, making it more difficult to give a correct diagnose. Others do not concentrate as much as wanted in the areas of osteomyelitis. Short half-lives and/or complicated chemistry make yet other radiotracers difficult to handle and to synthesize—in time or at all. For these reasons, new and better radiotracers should be developed. In such development, the radiochemist is one of several important players. If the tracer molecule cannot be synthesized, or if the radioactive element cannot be incorporated within the time limits set by the radioactive half-life, then there will be no radiotracer. 

In this paper, we will introduce a series of relevant radionuclides (Section 2) and then review radiotracers for diagnosing osteomyelitis. The review will include radiotracers in common use for the diagnosis (Section 3), as well as promising new radiotracers (Section 4). To illustrate that things are not always simple, we also include examples of theoretically promising tracers that did not live up to the expectations. As practical illustrations, we will include examples from a major project of our own, the porcine osteomyelitis project, in which haematogenous osteomyelitis was induced in the right hind limb of juvenile pigs (leaving the left hind limb uninfected). As demonstrated by Johansen et al. [4], a juvenile pig model mimics osteomyelitis in children. Throughout the review, emphasis will be put on the chemical and physiological mechanisms that make a given tracer interesting.

## 2. Radionuclides in Nuclear Medicine Imaging

This section contains a brief introduction to radionuclides and imaging by SPECT and PET. Readers well versed in these topics may skip this section and continue with Section 3.

To be relevant for in vivo imaging, a radionuclide must emit a kind of radiation that can be detected outside the body. This basically means that the decay must result in gamma-emission, either gamma-rays emitted from the nucleus or gamma-rays produced by a secondary process outside the nucleus. As detailed in the following, the former decays are used for gamma-camera/SPECT imaging, while the latter decays are used for PET imaging.

Gamma-rays emitted from the nucleus are the cornerstone of the so-called traditional nuclear medicine. The most important radionuclide in this context is technetium-99 in the excited state ^99m^Tc. When ^99m^Tc decays to the ground state ^99^Tc, a photon of energy 140 keV is emitted. The gamma camera measures these photons individually, giving an image of the distribution of ^99m^Tc (or other radionuclides) in the body. A gamma camera able to measure from a range of angles around the patient, thereby allowing tomographic imaging, is often referred to as a SPECT scanner (single-photon emission computed tomography). The half-life of ^99m^Tc is 6 h (see Table 1), suitable for the study of physiological processes happening on a time scale up to many hours, but not days. In nuclear medicine departments, ^99m^Tc can be obtained from technetium generators containing radioactive molybdenum decaying to technetium: ^99^Mo → ^99m^Tc + e^−^. Chemically, technetium is a metal which in water forms the pertechnetate ion, TcO_4_^−^. When the generator is rinsed (eluted) with saline water, the pertechnetate ion exchanges with Cl^−^ ions. As a result, ^99m^Tc is eluted, while ^99^Mo remains in the generator. A wide range of tracer molecules based on ^99m^Tc exists. 

Positron emission tomography (PET) is based on positron emission (β^+^ decay). A positron, e^+^, is the anti-particle of a normal electron, e^−^, and upon meeting the two can annihilate, a process in which the mass of both particles is converted into energy (E = mc^2^) in the form of two gamma rays: e^+^ + e^−^ → γ + γ. The two gamma rays fly in opposite directions, and the PET scanner measures these photon pairs. The PET image is computed from a large number of such measurement pairs (co-incidences), depicting the distribution of positron-emitting radionuclides in the patient. A very important radionuclide for PET imaging is fluorine-18, which can be produced in large amounts (activities) at hospitals with a cyclotron. As fluorine is a chemically very reactive element, ^18^F can be incorporated into a range of molecules. The half-life of ^18^F is 110 min (see Table 1), long enough to perform many kinds of syntheses and for studying physiological processes for up to a few hours.

As a technique, PET has advantages over SPECT, basically because detection of a pair of photons gives more (or more easily available) spatial information than detection of a single photon. This leads to better spatial resolution and scanners with higher detection efficiency, which roughly speaking transfers to better images.

An important aspect of radionuclides, apart from the kind of emission, is the physical half-life. The half-lives involved in nuclear medicine vary from minutes to days (see Table 1). A short half-life has the advantage that the patient will only be irradiated for a short period of time, for which reason the radiation dose will be low. This allows the injection of relatively high radiotracer activities, facilitating the acquisition of images of high quality. A longer half-life has the advantages that there will be more time available for the chemical synthesis of the radiotracer, there may be time for transport from one site to another, and longer-lived tracers can be used to study slower physiological processes. In relation to osteomyelitis, a very short-lived radionuclide, such as ^15^O, can be used in the form of radioactive water to study blood flow in the infected tissue, while it is useless for studying physiological processes happening over hours. A relatively long-lived radionuclide like ^111^In can be imaged after many hours or even a few days, allowing imaging of slower processes such as leukocyte concentration in infection; however, to limit the patient radiation dose, only a small activity of ^111^In can be used, which reduces image quality. Both of these examples are further detailed below.

## 3. Established Radiotracers for Bone Infection

A number of radiotracers are already in use for the diagnosis of infection in general and osteomyelitis in particular. For gamma-camera imaging of osteomyelitis, [^67^Ga]Ga-citrate and labelled leukocytes have been widely used. For PET scanning of osteomyelitis (and for PET scanning in general), [^18^F]FDG has a dominant role. Below is given a review of these tracers, including their uptake mechanism and their pros and cons. 

### 3.1. Gallium-Citrate (Based on Ga-67)

Originally used as a tracer for scintigraphy of malignant tumours, [^67^Ga]Ga-citrate was in 1971 recognized as showing uptake in infection [6]. While [^67^Ga]Ga-citrate has previously been an important tracer for infection/inflammation imaging, its use has now largely been taken over by labelled leukocytes (see below) [7,8]. Despite ^67^Ga being seldomly used today, it is of interest to study its uptake mechanism, both as an illustrative example and because the radioisotope ^68^Ga reviewed below has the same chemistry since it is a different isotope of the same chemical element.

Gallium (Ga) is a metal in group III of the periodic system, and its chemistry has some resemblance to that of iron (Fe). Similar to other metals, gallium is relatively insoluble at normal pH, while the complex Ga-citrate is soluble [7]. When Ga-citrate is injected into the blood, the complex immediately dissociates, and more than 99% of the Ga-ions associate with transferrin, a plasma protein normally transporting iron, while the rest, <1%, is taken up by leukocytes [9]. Thus, while [^67^Ga]Ga-citrate is injected, the actual tracer is [^67^Ga]Ga-transferrin. A major difference between gallium and iron is the inability of gallium to be reduced in vivo; while ferric ions are reduced and become part of haem, gallium remains bound to its carrier molecule (transferrin) [10]. The transfer of iron involves the reduction of Fe(III) to Fe(II), while gallium is not reduced from Ga(III), and the elimination rate is far slower for gallium than for iron [11]. 

Several theories exist regarding the uptake mechanism of gallium in tissues. Focussing on the inflammatory uptake, gallium may accumulate due to: [9,10,12]
Increased vascular permeability of the inflammatory tissue, allowing transferrin proteins (and thereby [^67^Ga]Ga-transferrin) to accumulate in the tissue.Uptake by leukocytes, where gallium binds to lactoferrin (another iron-binding protein). Leukocytes accumulate in the inflammatory tissue.Binding by siderophores (high-affinity iron-chelators), which are produced by bacteria in high amounts in iron-poor environments, such as inflammatory processes, to help the bacteria obtain sufficient iron.Binding by mucopolysaccharides found in the intercellular space of inflammatory sites.While the Ga-transferrin complex has high stability at normal pH in the blood, Ga dissociates from transferrin in environments of lower pH, as is often the case in tumours and abscesses.

Once taken up by the tissue, gallium is only slowly excreted, especially after the first 24 h [10]. Imaging with [^67^Ga]Ga-citrate is typically performed 1–3 days after injection [13]. Normal physiological uptake is seen in the liver, skeleton, lacrimal glands, salivary glands, and lactating breasts [12]. 

Compared to other radionuclides used for medical imaging, ^67^Ga has a relatively long physical half-life (Table 1), which results in a relatively high radiation dose to the patient for a diagnostic procedure. A guideline on gallium scintigraphy in inflammation [13] recommends a [^67^Ga]Ga-citrate activity of 150–220 MBq (or 4–6 mCi) for adults, which will give the patient a radiation dose of ~20 mSv. For comparison, a typical radiation dose from a whole-body CT scan of full diagnostic quality is typically ~15 mSv, and newer CT scanners may give less. Accordingly, the first paragraph of the gallium scintigraphy guideline states: “Alternative techniques, such as labelled leukocytes, should be considered if clinically indicated.”

(For gallium-citrate based on the isotope ^68^Ga, see Section 4.1).

### 3.2. Labelled Leukocytes

Labelled autologous leukocytes (white blood cells, WBC) are the conventional radionuclide gold standard for infection imaging [3,7,14]. Being part of the body’s immune system, leukocytes by nature seek sites of infection, including osteomyelitis. 

For gamma-camera imaging (scintigraphy or SPECT), leukocytes can be labelled with ^111^In or ^99m^Tc, which are taken up by the leukocytes: [^111^In]In-leukocytes or [^99m^Tc]Tc-leukocytes. Of these two, labelling with ^111^In is most stable as ^99m^Tc is eluted from the leukocytes to a higher extent than ^111^In. Furthermore, the 2.8-day half-life of ^111^In allows imaging the next day, giving time for leukocyte accumulation at sites of infection. In cases of uncertain findings, imaging can be repeated on day two and/or day three. Late imaging reduces background uptake at the cost of being logistically cumbersome. Note that late imaging does not increase the radiation dose to the patient, as it is imaging of the remains of the originally injected radiotracer, not a new activity. Normal physiological uptake is seen in the liver, spleen, and bone marrow [3,14,15].

Especially because of the long half-life, ^111^In-labelled leukocytes result in a relatively high radiation dose per MBq injected, requiring the use of relatively low activity in human patients. For that reason, the SNM guideline on ^111^In-labelled leukocytes [15] recommends ^99m^Tc-labelled leukocytes as an alternative for some clinical settings, particularly regarding paediatric patients. Labelling leukocytes with [^18^F]FDG has been investigated, as this would allow PET imaging. However, according to Palestro [8], it is unlikely that this will enter clinical practice: Labelling efficiency is significantly lower than for ^99m^Tc- and ^111^In-labelling, the 110-min half-life of ^18^F is short compared to the leukocyte infiltration time, and [^18^F]FDG is eluted from the leukocyte cells.

Whatever the radionuclide, in vitro leukocyte labelling is an elaborate process, requiring trained personnel. The procedure involves drawing of large blood sample (typically 40–80 mL), separation of the leukocytes, and then the actual radio-labelling of the separated leukocytes. This is followed by re-injection of the labelled leukocytes in the same patient or animal (autologous injection), waiting time for uptake, and only then the scan. These procedures are cumbersome and, at the same time, involve hygienic risks [15].

In vivo labelling of leukocytes has been attempted. The idea is to inject a radiotracer that is (selectively) taken up by leukocytes. This approach would circumvent the issues of in vitro labelling. ^99m^Tc-labelled murine antibodies or antibody fragments have been utilized for this purpose. Unfortunately, there have been cases of allergic reactions to the antibodies (applied as, e.g., 125 µg doses, including both the labelled and the unlabelled molecules), and marketed tracers of this kind have been withdrawn and/or are not widely available [3].

(For the use of signalling molecules related to leukocytes, see Section 4.7).

### 3.3. Bone Marrow Tracers (Labelled Nanocolloids)

Leukocytes accumulating in the natural reservoirs of healthy red bone marrow can lead to false positives in the interpretation of leukocyte scans. This is especially a problem when prosthesis-related osteomyelitis is suspected, as marrow distribution is often non-normal around a prosthesis. For example, the femoral bone marrow will be displaced from its natural distribution by the insertion of a hip prosthesis.

To facilitate interpretation of the images, leukocyte imaging is often combined with imaging of a bone marrow tracer without infection-specific uptake. Due to the different photon energies of ^111^In and ^99m^Tc, ^111^In-labelled leukocytes can be imaged simultaneously with a ^99m^Tc-labelled bone marrow tracer (dual-isotope scanning). Infection is indicated by the accumulation of leukocytes in locations without marrow (uptake seen only in the ^111^In image), while leukocyte accumulation at locations that contain bone marrow (uptake seen in both the ^111^In and the ^99m^Tc images) is normal [15].

Bone marrow tracers are usually colloidal nanoparticles of size below 100 nm, typically labelled with ^99m^Tc for scintigraphy or SPECT imaging. The particles are taken up by the liver, spleen, and bone marrow, i.e., corresponding to the normal physiological leukocyte uptake [16].

### 3.4. Bone Scintigraphy (Labelled Phosphonates)

The destruction of bone in osteomyelitis causes the body to attempt to build the affected bone. This can be imaged with bone scintigraphy (and bone SPECT) using ^99m^Tc-labelled diphosphonates (bisphosphonates). The uptake reflects blood flow and the rate of new bone formation, and bone scintigraphy is frequently used in the diagnosis of osteomyelitis. Imaging is, however, not specific for osteomyelitis, as uptake will be elevated where the body is replacing degraded bone, which happens in both bone tumours and bone infections, as well as in bone healing after a fracture [8].

Bone scintigraphy can be performed as a 3-phase bone scan: [8]
First phase: Dynamic imaging right after injection (e.g., the first minute), reflecting the blood perfusion.Second phase: Static imaging of the relevant part of the body immediately after the first phase, reflecting the blood pool.Third phase: After 2–4 h, static imaging, reflecting uptake of phosphonates in the bone matrix.

A typical phosphonate for bone scintigraphy is methylene diphosphonate (MDP). Others are hydroxymethane diphosphonate (HDP) and 2,3-dicarboxypropane-1,1-diphosphonate (DPD). The clinical differences between these three diphosphonates (bisphosphanates) appear to be minor [17].

The uptake mechanism is based on the diphosphonate complex coordinating with calcium in hydroxyapatite, which constitutes a major part of the bone matrix; however, not all details are clear. Historically, the first diphosphonate was pyrophosphate with a P-O-P structure. This tracer was later replaced by diphosphates with a P-C-P structure, e.g., MDP. New diphosphonates are being developed, in which the diphosphonate group is spatially more separated from the technetium-chelator. For a review, see [18].

For completeness, it should be mentioned that bone scans are increasingly being performed as PET scans using [^18^F]fluoride (^18^F^−^) as the tracer, although most applications are oncological to test for bone metastases. The fluoride is administered as a sodium fluoride solution, [^18^F]NaF. In the bone, the fluoride anions are exchanged with the hydroxyl groups in the hydroxyapatite (Ca_10_(PO_4_)_6_(OH)_2_), forming fluorapatite (Ca_10_(PO_4_)_6_F_2_) [18,19]. Our porcine osteomyelitis project included pigs scanned with [^18^F]NaF, but while bone growth zones in the juvenile pigs were very visible, osteomyelitis-specific uptake of this tracer was not seen [20].

### 3.5. FDG, a Glucose Analogue

Natural glucose has right-handed chirality, D-glucose. Left-handed chirality, L-glucose, can be formed but is not the biologically natural form. Figure 1 illustrates natural glucose along with two other molecules: 2-deoxy-D-glucose (DG) and 2-fluoro-2-deoxy-D-glucose (FDG).

Both DG and FDG are taken up by cells by the cells’ glucose transporters (GLUTs), but their metabolite products are trapped in the cell. More precisely, glucose, DG, and FDG are all phosphorylated by the glucose-metabolizing enzyme hexokinase. However, while the metabolite product of glucose (glucose-6-phosphate) is further metabolized, the metabolite products of DG (DG-6-phosphate) and FDG (FDG-6-phosphate) are not substrates of any enzymes. Furthermore, the cell membrane is very poorly permeable to these intermediate metabolite products. Thus, for both [^14^C]DG (for ex vivo radiographic imaging) and [^18^F]FDG (for in vivo PET imaging), the radionuclide is essentially trapped within the cell. This uptake and trapping of the tracer allow imaging of the glucose metabolism [21,22,23].

Physiologically, glucose has high uptake in the brain, working muscle tissue, and brown adipose tissue (brown fat) during hypothermia; brown adipose tissue is present mostly in small children but also in adults living in cold environments [24]. Many types of cancer tissue show high glucose uptake (the Warburg effect [23]), for which reason [^18^F]FDG is widely used for cancer diagnosis. However, the uptake is also elevated in both acute and chronic inflammation due to increased expression of GLUTs in the tissue [25].

Static [^18^F]FDG PET is typically performed after a waiting time of 60 min to allow for uptake and for the excretion of non-trapped tracer to reduce the background signal.

A meta-analysis on nuclear imaging for classic fever of unknown origin (FOU) [26] found [^18^F]FDG PET/CT to have overall 86% diagnostic sensitivity (95% confidence interval 81–90%), but only 52% (36–67%) diagnostic specificity; despite having lower specificity than gallium scintigraphy and leukocyte scintigraphy, the meta-analysis found [^18^F]FDG PET/CT to the have best overall performance. A review on radionuclide imaging of osteomyelitis [8] calls [^18^F]FDG PET/CT “extremely useful in the diagnostic workup of osteomyelitis,” with reported sensitivity >95% and specificity 75–99% (although based on fewer studies than the FOU meta-analysis).

Within our porcine osteomyelitis project, static imaging found [^18^F]FDG to be superior to other tracers for marking lesions but also with unspecific uptake [27,28,29].

The main disadvantage of [^18^F]FDG is its lack of specificity. Its uptake reflects the sugar uptake, which is elevated not only in infection but also in inflammation, tumours, the brain, and striated muscles (most when a muscle is working). The physiological uptake can give rise to false positives, and foci right next to tissues of high physiological uptake (e.g., skull osteomyelitis vs. natural brain uptake) may be difficult to differentiate. For the body in general, diagnosis is hampered by the lack of differentiation between inflammation, infection, and tumour. Some of the osteomyelitis foci seen in the clinical everyday life are caused by joint replacements, and other infections are caused by insertion of stents or arterial replacement. These surgical procedures will often cause inflammation, so until healing is complete, an [^18^F]FDG scan will show uptake at the place, whether or not the intervention caused an infection.

## 4. Newer Tracers Considered for Bone Infection Imaging

In recent years, a number of other tracers have been investigated for infection imaging generally and bone infection specifically. Many of these are PET tracers (i.e., incorporating positron-emitting radionuclides). The focus on PET is largely because PET imaging as a technique is advantaged, as described earlier. Tracers based on ^99m^Tc, the workhorse of traditional nuclear medicine, are, however, also represented. Our main focus is bone infections. For a broad focus, we refer the readers to the recent and well-written review of new infection tracers by Welling et al. [30].

### 4.1. [^68^Ga]Ga-Citrate

As described above, infection imaging based on gallium-citrate based on ^67^Ga was introduced half a century ago but is not used very much today.

As reviewed by Kumar and Boddeti [31], the replacement of the SPECT radionuclide ^67^Ga by the PET radionuclide ^68^Ga can combine the technical advantages of PET/CT over SPECT/CT (higher measurement sensitivity, higher image quality, and quantitative imaging) with a reduced patient dose due to the shorter half-life of the radionuclide. Furthermore, ^68^Ga can be produced with a ^68^Ge/^68^Ga-generator, which makes ^68^Ga accessible for many nuclear medicine departments without direct access to ^67^Ga. The shorter half-life of ^68^Ga requires imaging to be performed within hours rather than days, resulting in a visible vascular signal but avoiding a several-day patient protocol. [^68^Ga]Ga-citrate can be produced by a very fast and simple synthesis [32].

Mäkinen et al. [33] compared [^18^F]FDG and ^68^Ga in a rat model involving both osteomyelitis and infection-free bone healing. Relative to healthy bone, the study found both tracers to accumulate in osteomyelitis, but in contrast to [^18^F]FDG, ^68^Ga did not accumulate in infection-free healing bone, indicating the possibility of a tracer able to distinguish infection from inflammation. The authors noted as a shortcoming that the rat osteomyelitis model required injection of a large inoculum directly into the bone, for which reason the model clinically “perhaps best simulates osteomyelitis arising from grossly contaminated long-bone fractures” [33].

Nanni et al. [34] used [^68^Ga]Ga-citrate in 31 patients suspected of acute osteomyelitis, chronic osteomyelitis, or diskitis. Diagnostic results for detecting infection (any of the indications) had 100% sensitivity (23 positives of 23) and 76% specificity (13 negatives of 17). The results were considered preliminary, partly due to a lack of direct comparison with [^18^F]FDG.

In our own porcine osteomyelitis study, uptake of [^68^Ga]Ga-citrate was found inferior to that of [^18^F]FDG, especially for bone lesions, less so for soft tissue lesions [29,35].

### 4.2. Other Sugar Analogues Than FDG

As tracers of bacterial infection, sugar analogues, in general, have the advantage that sugars are taken up by the cell volume and, for this reason, can reach higher concentrations than are typically possible for tracers that bind to the cell surface only [36]. As described above, the glucose analogue [^18^F]FDG is a very sensitive infection tracer but is not specific for infection. Sugar analogues with a more selective uptake mechanism than FDG can show more specific uptake.

An example is 2-[^18^F]fluoro-2-deoxy-sorbitol ([^18^F]FDS), a PET-tracer analogue of sorbitol. [^18^F]FDS is metabolized by Enterobacteriaceae (e.g., *E. coli*) but not in Gram-positive bacteria (e.g., *S. aureus*) or healthy mammalian or cancer cells in vitro. In a murine myositis model, PET scanning with [^18^F]FDS rapidly differentiated true infection from sterile inflammation with a limit of detection of 6.2 ± 0.2 log_10_ colony-forming units (CFU) for *E. coli* [37]. [^18^F]FDS can be simply produced from [^18^F]FDG by reducing the aldehyde group to a hydroxyl group, using NaBH_4_ [38]. The usefulness of [^18^F]FDS as a PET tracer to image Enterobacteriaceae-specific infections has also been evaluated in humans. The authors conclude that [^18^F]FDS PET holds great potential for appropriate and effective for the imaging of bacterial infections in vivo, including the ability to distinguish Gram-negative infections from Gram-positive infections [39].

Other tracers target the sugar transport system. In multicellular organisms, some cells specialize in providing sugars to other cells (e.g., intestinal and liver cells in animals, photosynthetic cells in plants), whereas others depend completely on an external supply (e.g., brain cells, roots, and seeds) [40]. Therefore, the transport and uptake of sugar are very essential for multicellular organisms to work. Most sugars are derived from photosynthetic plants, and for some sugars, the uptake is different in animals and bacteria. The Sugar Phosphotransferase System (PTS) is only found in bacteria, and the ABC (ATP-binding cassette) transporter proteins appear to be involved in sugar transport only in bacteria [40]. The right choice of sugar may have a high uptake in bacteria (infection) while having a very low uptake from animal/human physiology (non-infected tissue and organs).

As a specific example, maltose and maltodextrin are taken up by the *E. coli* maltose ABC transporter [41]. This was explored by researchers who synthesized maltodextrin-based imaging probes. They shoved that maltodextrin was internalized through the bacteria-specific maltodextrin transport pathway and that these tracers can be used to detect bacteria in vivo with high sensitivity and specificity based on this bacteria-specific mechanism [36,42]. In vitro results on bacterial biofilms showed that high and specific uptake was not limited to *E. coli* but could be demonstrated for biofilms from a wide range of bacterial families, including the *S. aureus* species. The future will show to what extent maltodextrin tracers will become widespread and to what extent they could be used for OM detection.

### 4.3. [^15^O]water

For most radiotracers, the supply of the tracer to a given tissue will be influenced by the blood perfusion of the tissue; the exceptions being tracers where uptake is not limited by the supply of tracer but by the difficulty of diffusion into the tissue. For inflamed tissue, perfusion is most often elevated. For bone tissue in osteomyelitis, however, this appears to be less pronounced [43].

Uptake and release of [^15^O]water, or [^15^O]H_2_O, are generally modelled with a 1-tissue compartment model (1TCM), i.e., a model with input from the blood to a single tissue compartment—unlike, e.g., FDG, nothing further happens to water in tissue (except for returning to the blood pool). Water passes capillary walls with ease, and the first-pass extraction fraction (*E*) in tissue is close to 100% [44]. Assuming *E* = 100%, perfusion will equal the immediate rate of uptake.

The half-life of ^15^O is only two minutes, limiting the time available for scanning to a few minutes but also ensuring a low radiation dose (relevant for human studies). Furthermore, only a short waiting time (10–15 min) is needed before a new scan can be made.

Quantitative measurement of blood perfusion with PET scanning using [^15^O]water as a tracer has been performed since the early 1980s [45,46]. Today, the method is an established method for quantitative in vivo measurement of blood perfusion, but apparently, our porcine osteomyelitis project was the first to use [^15^O]water in relation to osteomyelitis [43].

### 4.4. [^11^C]methionine

Methionine is an essential amino acid necessary for the normal growth and development of animals and humans [47]. The PET tracer [^11^C]methionine is chemically identical to natural methionine, incorporating ^11^C into the CH_3_ group on the sulphur atom (Figure 2). Methionine is metabolized in the body through a complex series of processes [47]. If not incorporated into protein, [^11^C]methionine can undergo transfer to adenosyl-[^11^C]methionine, followed by loss of the [^11^C]methyl group to a variety of methyl acceptor enzymes. For this reason, both tissue and plasma may contain some radioactive metabolite products.

Methionine uptake reflects increased amino acid transport and protein synthesis and is related to cellular proliferation [49]. Protein synthesis and cellular proliferation are up-regulated in infected tissue, and increased [^11^C]methionine uptake has been reported in brain infections [50,51]. Within the porcine osteomyelitis project, [^11^C]methionine was on static imaging found to visibly accumulate in 6 of 10 infectious lesions in pigs no. 1–4 [29] and 19 of 24 osteomyelitis foci in pigs no. 6–10 [27].

Stereochemistry appears to play an unexpected role in this context. Most amino acids exist in two chiral forms, L and D, but animal and human physiology are homochiral, being based on the L-form [52]. This is broadly known, and accordingly, the synthesis of [^11^C]methionine for PET scanning is normally based on L-methionine; all the above-cited PET studies on [^11^C]methionine used L-methionine.

A 2017 study has, however, demonstrated that both *E. coli* and *S. aureus* have high uptake of D-methionine, while the D-form has low uptake in animal and human tissue, for which reason D-[^11^C]methionine in a mouse model was demonstrated to have much more infection-specific uptake than L-[^11^C]methionine [53]. The group has since then developed a new synthesis for D-[^11^C]methionine, performed in vitro tests showing uptake of the tracer in a broad range of clinically relevant bacteria, and are preparing for in-human evaluation of D-[^11^C]methionine in patients with, e.g., vertebral discitis-osteomyelitis [54].

In retrospect, uptake of D-form methionine (and other amino acids in the D-form) in bacteria should perhaps not be surprising. Such uptake is in accordance with Friedman [52], who mentions that D-amino acids are generally part of bacterial cell walls, which contributes to their resistance to digestion by the body’s (L-form) proteolytic enzymes. It will be interesting to see how the potential of D-form amino acids as infection tracers unfolds in the coming years.

### 4.5. [^11^C]donepezil

Donepezil (also known as donepezil hydrochloride, E2020, or Aricept) is a reversible inhibitor of acetylcholinesterase (AChE), an enzyme breaking down the neurotransmitter acetylcholine (ACh). Donepezil was developed as a treatment drug to alleviate the cognitive impairment caused by Alzheimer’s disease (AD) [55].

Figure 3 shows [^11^C]donepezil labelled with ^11^C as part of the methoxy (O-CH_3_) group on the 5th carbon atom. The short wording “[^11^C]donepezil” will here denote [5-^11^C-*methoxy*]donepezil.

The PET tracer [^11^C]donepezil is generally used as a PET tracer for brain scans of AD patients. However, it has been demonstrated that [^11^C]donepezil also has infection-related uptake [57,58]. This apparently reflects that ACh is not only a neurotransmitter but also has a regulatory role in immune cells, which therefore express AChE [59,60].

Jørgensen et al. [58] studied the uptake of [^11^C]donepezil (and [^18^F]FEOBV and [^18^F]FDG) in infection and inflammation in mice, pigs and humans. In all three species, [^11^C]donepezil showed elevated uptake in infectious regions; in one human, asymptomatic pneumonia had a particularly high uptake.

Afzelius et al. [27] studied the uptake of [^11^C]donepezil in five pigs, based on static PET/CT imaging performed one hour post-injection (p.i.). Diagnostically visible uptake was found in 14 of 24 OM lesions and 4 of 4 abscesses, and 6 of 8 enlarged lymph nodes. It was concluded that while [^11^C]donepezil had some ability to detect osteomyelitis, the accumulation was too weak for the tracer to be suitable for this purpose.

The evaluation of blood samples and dynamic PET data from the same project indicated a faster metabolism of [^11^C]donepezil in pigs than in humans; the analyses were complicated by the fact that an important metabolite product included the ^11^C atom (thus being visible on the PET scans) and had an affinity for the same receptors as donepezil [35]. These findings exemplify that for radiotracers with non-negligible metabolism during the studying time, the kinetics of the radioactive metabolites must be considered.

### 4.6. Technetium-Labelled Interleukin-8 (IL-8)

In the blood of most mammals, neutrophilic granulocytes are the most frequent type of white blood cells (40–75%). They are an essential part of the innate immune system. Neutrophilic granulocytes are phagocytic cells. At the beginning of a bacterial infection, neutrophilic granulocytes are early responders to infections and migrate through the blood vessels and the interstitial tissue towards the site of infection, following chemical signals, such as interleukin-8 (IL-8), in a process called chemotaxis. The neutrophilic granulocytes have high expression of 2 types of interleukin-8 (IL-8) receptors, CXCR1 and CXCR2 [61], and IL-8 binds to the receptors with high affinity [62,63]. Lauersen et al. [64] have demonstrated by immunohistochemistry on porcine samples that IL-8 is a predominantly neutrophil-related cytokine but that it is also present in epithelial and blood vessel endothelial cells of the inflammatory process.

Rennen et al. [65] showed that IL-8 could be labelled with ^99m^Tc using HYNIC as a chelator with preservation of its leukocyte receptor binding capacity in rabbits with *E. coli* infection. The preparation allows visualization of the infection with high target-to-background ratios within a few hours after injection.

In a rabbit model of osteomyelitis, Gratz et al. [66] found that [^99m^Tc]Tc-IL-8 clearly revealed the osteomyelitis lesions. Although the absolute uptake in the osteomyelitis area was significantly lower than the uptake obtained with [^99m^Tc]Tc-MDP and [^67^Ga]Ga-citrate, the target-to-background ratios were significantly higher for [^99m^Tc]Tc-IL-8 because of fast background clearance.

The first clinical evaluation of [^99m^Tc]Tc-IL-8 scintigraphy demonstrated that injection of the tracer was well tolerated and allowed the detection of various infections in patients at 4 h after tracer injection. Notably, the tracer accumulated in osteomyelitis but not in non-infectious disorders, such as joint prostheses showing aseptic loosening. The radiation burden from the injection of 400 MBq of [^99m^Tc]Tc-IL-8 was low, typically 2.7 mSv [67].

The porcine osteomyelitis project found that the tracer has good potential for the detection of osteomyelitis (OM), detecting approximately 80% of OM lesions (imaging time had some variation, 3–6 h p.i.) at the same level as ^111^In-labeled autologous leukocytes [68].

### 4.7. [^68^Ga]Ga-DOTA-Siglec-9

As reviewed in Section 3.2, it is possible to label leukocytes, which are natural seekers of infection, but the process is cumbersome. A different approach is to target some of the signalling molecules involved in the process.

Vascular adhesion protein-1 (VAP-1) is a protein involved in the process of leukocyte adhesion at infected sites and, at the same time, functions as a regulatory enzyme for leukocyte binding at sites of infection and inflammation [69,70,71]. Sialic acid-binding immunoglobulin-like lectin 9 (Siglec-9) is one of the body’s natural ligands for VAP-1, which has attracted attention as a tracer molecule for infection imaging [72,73,74]. More precisely, a ^68^Ga-labelled modified fragment of natural Siglec-9 is used as a PET tracer, [^68^Ga]Ga-DOTA-Siglec-9. For an overview of the development and details on the radiosynthesis of the tracer, see [71]. Focussing on bone infection, the accumulation at osteomyelitis sites has been demonstrated, although not to the extent of [^18^F]FDG accumulation [75].

[^68^Ga]Ga-DOTA-Siglec-9 has recently been tested in humans, finding the application safe and demonstrating that the tracer could image rheumatoid arthritis with the uptake visually comparable to uptake of [^18^F]FDG [76].

### 4.8. ^68^Ga-Labelled Phage-Display Selected Peptides

Our own group set out to develop a ^68^Ga-labelled PET tracer aiming to target an *S. aureus* biofilm. A biofilm was grown in vitro from an *S. aureus* sample obtained from an infected knee joint of a patient. We applied phage-display to select peptide sequences with an affinity for the biofilm, and three promising peptide sequences (A8, A9, and A11) were selected and synthesized with a DOTA chelator affiliated to the peptide sequence via a lysine linker. One sequence (A8) was discarded due to instability, leaving two as potential infection tracers (A9 and A11). The two tracers were named [^68^Ga]Ga-DOTA-K-A9 and [^68^Ga]Ga-DOTA-GSGK-A11 after they had been labelled with gallium-68. In vitro studies showed that these potential tracers had an affinity for *S. aureus* as well as other bacteria [77].

[^68^Ga]Ga-DOTA-K-A9 was tested in a mouse model, where an *S. aureus* culture was subcutaneously injected into mice. It was found that [^68^Ga]Ga-DOTA-K-A9 accumulated at the sites of infection but also to some extent at sterile inflammations. Moreover, in vitro experiments suggested binding to an intracellular epitope, which may be the result of the tracer targeting the compromised cell membranes of dead bacteria [78]. Furthermore, both [^68^Ga]Ga-DOTA-K-A9 and [^68^Ga]Ga-DOTA-GSGK-A11 were tested in a pig model, the result being quite disappointing as no tracer accumulation was found in the sites of osteomyelitis [20].

In retrospect, a possible explanation for these results can be that the phage-display selected peptides with affinity for dead bacteria in the *S. aureus* biofilm instead of the intended selection of living bacteria. Knowing what we know now, the initial selection should have been followed by a different phage display targeted to deselect against dead bacteria. This procedure would have ensured that the chosen peptides had an affinity for living rather than dead bacteria, which would have increased the chance for the peptide to have a specific affinity for the *S. aureus* bacteria in infection.

### 4.9. Labelled Antimicrobial Peptides, Ubiquicidin

Differentiation between bacterial infection and sterile inflammation is of special interest. Antimicrobial peptides (AMPs) are part of the body’s natural defence against microbes. AMPs can distinguish between mammalian and bacterial or fungal cells and may be targeting vector candidates for molecular imaging due to their selectivity for bacterial cytoplasmic membranes in the innate immune system response [79]. AMPs are investigated in both therapy and diagnostics. The context here is diagnostics, and the focus will be on the specific AMP ubiquicidin.

Ubiquicidin was originally identified in mice, with homologue peptides being found in human tissue; its name was derived from the Latin ubique, meaning everywhere, due to its widespread presence [80]. Welling et al. [81,82] investigated ^99m^Tc-labelled fragments of the full peptide for scintigraphic imaging of infection, finding the fragment ubiquicidin (29-41) to have some of the highest target-to-nontarget (infected vs. non-infected) ratios across various bacterial species and to be able to differentiate between infection and sterile inflammation. Other fragments may have potential, too, but the (29-41) fragment has received the most attention; the (29-41) amino acid sequence is Thr-Gly-Arg-Ala-Lys-Arg-Arg-Met-Gln-Tyr-Asn-Arg-Arg (TGRAKRRMQYNRR), molecular weight 1693 Da. In the following, this fragment will be denoted UBI_29-41_.

For PET imaging, ubiquicidin fragments have been ^68^Ga-labelled with the NOTA chelator, as the chelation procedure requires less heating with NOTA than with DOTA, something that can be of importance to avoid damaging the peptides [79]. [^68^Ga]Ga-NOTA-UBI_29-41_ has been tested for safety and tested in humans, finding uptake in infection, including one case of osteomyelitis [83].

[^68^Ga]Ga-NOTA-UBI_29-41_ was also tested in the porcine osteomyelitis project but with no signs of uptake in infection. The reasons for the lack of uptake were not clear [20].

A good overview of clinical tests of [^99m^Tc]Tc-UBI_29-41_ and preclinical results for [^68^Ga]Ga-NOTA-UBI_29-41_ up to 2016 is found in a review by Ferro-Flores et al. [84].

## 5. Discussion

Radioactive tracers are very well suited for use in physiological imaging, as the radionuclide may be incorporated into molecules whose uptake is determined by the biochemistry and the physiology of the living body. This can be used in many contexts, including diagnostic of infection in general and osteomyelitis specifically. The uptake of a radiotracer can be due to its chemical properties (such as gallium), its ability to mimic naturally occurring molecules (such as sugar analogues), or by being a labelled variant of a naturally occurring cell or compound of the body (such as labelled leukocytes or methionine). Rather than directly imaging infection, radioactive tracers can also be used to visualize or measure processes, which are different in infected and non-infected tissue, such as blood flow. The emission of gamma rays makes it possible to visualize radiotracers in the living body, and a number of suitable radionuclides with short half-lives exist.

As demonstrated by this review, a very wide range of possibilities exist, each with its pros and cons. The probably most widely used infection tracer at this time is the glucose analogue [^18^F]FDG, which has a very high sensitivity for detecting infection, including osteomyelitis, but is not specific for infection. Labelled leukocytes are more specific, even though for osteomyelitis, their natural accumulation in bone marrow can be a pitfall, but leukocytes are very cumbersome to label, and the image quality is lower.

The ideal tracer would have high and specific uptake in infection, without uptake in other tissues or sterile inflammation. Furthermore, it should be simple to produce. The perfect tracer is unlikely to exist, but newer tracers may overcome many of the present shortcomings. Tracers built upon biologically relevant molecules related to either bacteria or the immune system’s response to microbes seem to us especially promising. This includes non-glucose sugar analogues targeted for bacteria rather than body tissue and antibodies, such as IL-8 or ubiquicidin.

The development of successful new tracers is not an easy task and requires a broad range of knowledge—broader than can be expected to find within a single profession. Medical doctors and veterinarians will often be those having the most knowledge about the physiology of infection, e.g., which proteins will accumulate in infection or which sugars are taken up specifically by bacteria. The synthesis of relevant analogues of these molecules involves general chemistry and biochemistry. Labelling with radionuclides involves physics and radiochemistry. If the potential bone marrow infection tracer makes it to preclinical testing, this will generally involve veterinarian issues (but those are beyond the scope of this review). In short, collaboration is important.

## 6. Conclusions

As demonstrated by the review, molecular radiotracers allow studies of physiological processes, including infection and osteomyelitis. The ideal tracer has not been found, and most likely will not, but the existing tracers nonetheless cover a wide spectrum, and some of the experimental tracers seem promising, with quite high and specific uptake and ability to distinguish infection from inflammation. Knowledge of uptake mechanisms as well as pitfalls and challenges is useful in both the use and the development of medically relevant radioactive tracers.

## Figures and Tables

**Figure 1 molecules-26-03159-f001:**
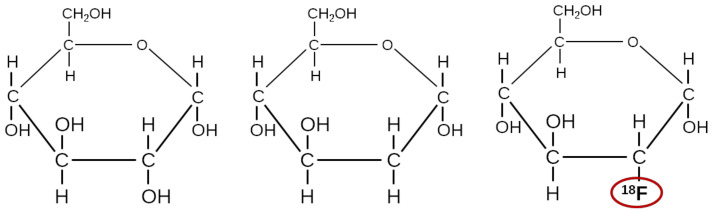
From left-to-right: natural glucose, DG, and [^18^F]FDG.

**Figure 2 molecules-26-03159-f002:**
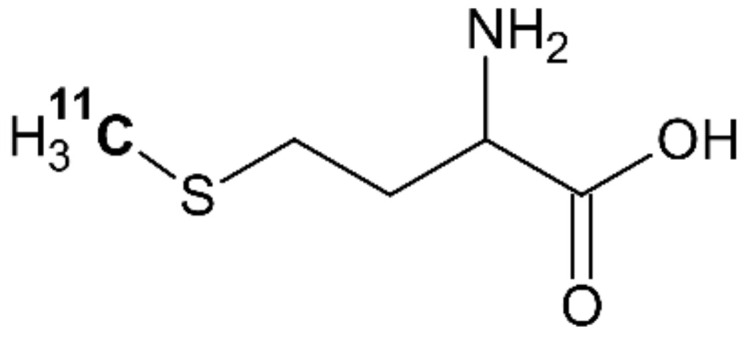
The chemical structure of [^11^C]methionine, redrawn from [48].

**Figure 3 molecules-26-03159-f003:**
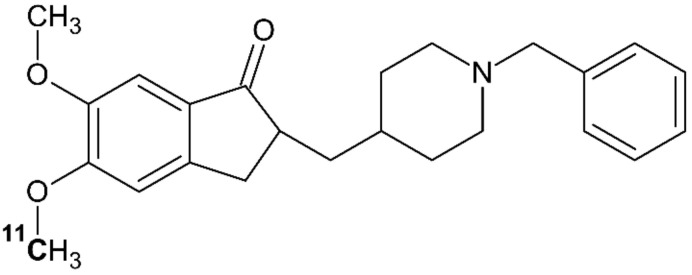
The chemical structure of [5-^11^C-*methoxy*]donepezil, redrawn from [56].

**Table 1 molecules-26-03159-t001:** An overview of radionuclides and their half-lives.

Radionuclide	Half-Life ^1^	Imaging Modality ^2^
^11^C	20.39 min	PET
^15^O	122 s	PET
^18^F	109.77 min	PET
^67^Ga	3.26 d	SPECT
^68^Ga	67.7 min	PET
^99m^Tc	6.02 h	SPECT
^111^In	2.80 d	SPECT

^1^ Source: ICRP Publication 107 [5] ^2^ PET: positron emission tomography. SPECT: single-photon emission computed tomography.

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
