# Peer review of "Radiotracers for Bone Marrow Infection Imaging"

_molecules, 2021, doi:10.3390/molecules26113159_

Round 1

Reviewer 1 Report

The manuscript is a well-prepared review on the use of the radiotracer technique for bone marrow infection imaging. Publication contains all published bone marrow infection imaging methods, both PET and SPECT. The publication is well prepared and should be published. 

I have some remarks to the manuscript:
chapter 2. "Radionuclides in nuclear medicine imaging" should be omitted. These are foundational information found in many textbooks. You just add reference.
line 61. The sentence "The use of short-lived radionuclides (half-lives from days down to minutes) ......" applies not only to bone radiopharmaceuticals but to all diagnostic radiopharmaceuticals.
line 171. The main mechanism of 67,68Ga accumulation in transferrin receptors was not mentioned.  Also the important difference between Fe3+ and Ga3+ has not been described (Ga has no oxidation state +2 ) that causes Ga3 + accumulation, iron in form Fe2+ leaves cell.

3.4. Bone scintigraphy (labelled phosphonates) point is too short. A mechanism for the accumulation of bisphosphonates can be added. Also labeled bisphosphonates of I, II and III generation  should be briefly described.

Author Response

We thank the reviewer for the positive overall evaluation, and for comments which have helped us strengthen the manuscript. Answers to specific comments are given below.

Comment: chapter 2. "Radionuclides in nuclear medicine imaging" should be omitted. These are foundational information found in many textbooks. You just add reference.

Answer: We agree that for readers from the field of nuclear medicine, this is basic knowledge, but we wrote this as a gentle introduction for those readers of “Molecule” who may read this special issue to learn about these techniques. In the revised version of the manuscript, the text is retained, but now starting with the information that readers may move directly to section 3, if they prefer.

Comment: line 61. The sentence "The use of short-lived radionuclides (half-lives from days down to minutes) ......" applies not only to bone radiopharmaceuticals but to all diagnostic radiopharmaceuticals.

Answer: True, this is a general statement. We have added the words “In general,” to make this clearer.

Comment: line 171. The main mechanism of 67,68Ga accumulation in transferrin receptors was not mentioned. Also the important difference between Fe3+ and Ga3+ has not been described (Ga has no oxidation state +2 ) that causes Ga3 + accumulation, iron in form Fe2+ leaves cell.

Answer: In the revised manuscript, the main mechanism has been more clearly described, and important differences between iron and gallium uptake have been added.

Comment: 3.4. Bone scintigraphy (labelled phosphonates) point is too short. A mechanism for the accumulation of bisphosphonates can be added. Also labeled bisphosphonates of I, II and III generation should be briefly described.

Answer: The uptake mechanism is now better described, and different generations of bisphosphonates are briefly described. A few words have also been added on NaF PET as bone scan, but not much as its main applications are in oncology and as bone scans in general are not specific for osteomyelitis.

Reviewer 2 Report

This is a nice review on the subject. Under current pandemic, this reviewer was expecting something related to viral infection and/or associated inflammation, but the narrow focus on osteomyelitis is fine so that all infection here are bacterial.

There are a few very minor points for consideration to improve the manuscript:

  1. Towards the end of 3.1, please mention that 4.1 will discuss the newer version of Ga-68 label for better connection;
  2. Towards the end of 3.2, after "99mTc-labelled murine antibodies or antibody fragments have been utilized for this purpose. Unfortunately, ..., and marketed tracers of this kind have been withdrawn and/or are not widely available. [3]", please mention 4.7 for new leukocyte targeting for better connection;
  3. For 4.5, readers are not sure whether [C-11]donepezil will be useful or not. Please be more definitive as possible.   

Author Response

We thank the reviewer for the positive evaluation and for comments which have helped us make the presentation more clear.

Comment 1.Towards the end of 3.1, please mention that 4.1 will discuss the newer version of Ga-68 label for better connection

Answer: This is now mentioned.

Comment 2.Towards the end of 3.2, after "99mTc-labelled murine antibodies or antibody fragments have been utilized for this purpose. Unfortunately, ..., and marketed tracers of this kind have been withdrawn and/or are not widely available. [3]", please mention 4.7 for new leukocyte targeting for better connection;

Answer: This is now mentioned.

Comment 3.For 4.5, readers are not sure whether [C-11]donepezil will be useful or not. Please be more definitive as possible.

Answer: We agree that this was unclear, and the text has been revised to be clear.